analytical chemistry

human plasma, meropenem, pharmaceutical formulation, tandem mass spectrometry, ultra-performance liquid chromatography, vaborbactam

**Author for correspondence:**
Maha A. Hegazy
e-mail: mahahgazy@yahoo.com

This article has been edited by the Royal Society of Chemistry, including the commissioning, peer review process and editorial aspects up to the point of acceptance.

# Ultra-performance liquid chromatography-tandem mass spectrometric method for quantitation of the recently Food and Drug Administration approved combination of vaborbactam and meropenem in human plasma

Ahmed K. Kammoun[1], Alaa Khedr[1], Ahdab N. Khayyat[1] and Maha A. Hegazy[2]

[1]Department of Pharmaceutical Chemistry, Faculty of Pharmacy, King Abdulaziz University, PO Box 80260, Jeddah 21589, Saudi Arabia
[2]Analytical Chemistry Department, Faculty of Pharmacy, Cairo University, Kasr El-Aini Street, 11562 Cairo, Egypt

MAH, 0000-0002-7486-1423

A parenteral medical combination containing vaborbactam and meropenem is used mainly to treat complicated urinary tract infections. A novel ultra-performance liquid chromatography tandem mass spectrometric method was developed for the sensitive determination of both compounds in human plasma. Sample preparation was performed by precipitation technique. The chromatographic separation was accomplished using the Acquity C18-BEH column, 0.01 M ammonium formate: acetonitrile (47:53, v/v) as a mobile phase with a flow rate of 0.2 ml min$^{-1}$. Analytes were monitored by applying multiple reaction monitoring. The bioanalytical validation criteria were conducted following the Food and Drug Administration recommendations. The method was linear within range 0.5 to 50 µg ml$^{-1}$, for both drugs. The intra-day and inter-day precision, as coefficient variation (% CV) and the accuracy, as % bias did not exceed 15% for both drugs. The percentage recovery of targeted analytes was not less than 77%, calculated at three quality control levels. The proposed method showed a suitable

lower level of quantification value of 0.50 µg ml$^{-1}$ for both analytes, which is far lower than the expected $C_{max}$, which permits the use of this method for pharmacokinetic studies. The proposed method proved to be useful for the evaluation of this combination in both human plasma and pharmaceutical formulation.

# 1. Introduction

Vaborbactam (VBR)–meropenem (MRP) has been recently approved by the Food and Drug Administration (FDA). It is first-in-class boronic acid-based β-lactamase inhibitor and a carbapenem combination product. This combination has been introduced for treatment of complicated urinary tract infections [1]. VBR is chemically designated as 2-[(3R,6S)-2-hydroxy-3-[(2-thiophen-2 yl-acetyl)amino]oxaborinan-6-yl]acetic acid. It has been tested for treatment of bacterial infections in renal failure patients [2]. Its β-lactamase inhibition is based on a cyclic boronic acid pharmacophore [2]. MRP, (4R,5S,6S)-3-[(3S,5S)-5-(dimethylcarbamoyl)pyrrolidin-3-yl]sulfanyl-6-[(1R)-1-hydroxyethyl]-4-methyl-7-oxo-1-azabicyclo[3.2.0]hept-2-ene-2-carboxylic acid is a broad spectrum carbapenem that is intravenously administered for severe bacterial infections [3]. The chemical structures for the studied drugs are presented in figure 1. In August 2017, the FDA approved a new intervention in the treatment of complicated urinary tract infections under the market name Vabomere® [4]. Vabomere® (MRP and VBR) for intravenous administration supplied as 2 g per vial: MRP 1 g/VBR 1 g. In Vabomere®, VBR is added to the therapy to improve the antibacterial effect by reducing the degree of MRP degradation by inhibiting the serine beta-lactamases. The treatment aims to resolve infection-related symptoms of complicated urinary tract infections [4].

Many chromatographic methods were reported for determination of MRP either singly or in combination with other drugs, including VBR [5–11]. Spectrophotometric methods have also been reported for the determination of MRP [12,13]. Parker *et al*. have developed a high performance liquid chromatography-mass spectrometry (HPLC-MS) method for monitoring VBR and MRP in human plasma and renal replacement therapy effluent [14]. The LC-tandem mass spectrometry (MS/MS) technique provides an information-rich tool for comprehensive identification and quantitation of drugs in different matrices [15–18]. Fast data-acquisition speed for the analysis and the highest sensitivity are leading benefits of this technique especially upon using ultra-performance liquid chromatography (UPLC)-MS/MS. Therefore, it was valuable to use UPLC-MS/MS for analysis of the studied novel combination in pharmaceutical formulation for help in batch release as well as to extend its application to human plasma for possible application for bioequivalence evaluation of generics versus the brand formulation, though, the development of such a method will be considered a pioneer and novel contribution in the bioanalytical determination of VBR and MRP. Also, the use of UPLC has enabled the use of a short column with a small particle size, which has resulted in well-defined chromatographic peaks within a very short run time. Thus, it is possible to analyse a large number of plasma samples per day.

Therefore, our aim in this work is to establish a highly selective bio-validated method for verification of the studied mixture in its dosage form and human plasma applying the UPLC-MS/MS technique, thus to allow its application for determination of the compounds for routine quality control (QC) works, in bioequivalence and bioavailability studies.

# 2. Experimental

## 2.1. Chemicals and reagents

VBR and MRP, 99.96%, reference standards were purchased from MedChemExpress® LLC, the USA and National organization of drug control and research, Egypt, respectively. The purity of the standards was 99.85% and 99.96%, in order, according to the manufacturer's certificate of analysis. All chemicals used throughout the work were of analytical grade, and all used solvents were of HPLC grade; acetonitrile and methanol were purchased from Merck, Darmstadt, Germany. Ammonium formate and formic acid were purchased from Sigma Aldrich, USA. Human plasma was obtained from the national vaccines and sera manufacturer (Vaccera), Giza, Egypt. The plasma samples were stored at −80°C.

| precursor ion (Q1) | product ion (Q3) |
|---|---|

**Figure 1.** LC-MS precursor (Q1) and product (Q3) ions of analysed analytes.

**Table 1.** UPLC-MS/MS parameters selected for quantitation of vaborbactam and meropenem using Ceftriaxone as an internal standard.

| compound[a] | Q1, m/z | Q3, m/z | CV, volt | CE, volt |
|---|---|---|---|---|
| vaborbactam | 298.16 | 96.89 | 18 | 34 |
| meropenem | 384.11 | 67.96 | 34 | 36 |
| ceftriaxone | 555.09 | 396.03 | 22 | 12 |

[a]Q1 is the precursor ion, Q3 is the product ion, CV is the cone voltage and CE is the collision energy.

## 2.2. Pharmaceutical formulations

VABOMERE® vial for intravenous injection is manufactured by Melinta Therapeutics Inc, the antibiotics company (East Coast, USA). This product is available as 2 g per vial: MRP 1 g/VBR 1 g without the addition of excipients and additives [2].

## 2.3. Chromatography

LC-MS/MS analysis was accomplished using a Xevo® triple stage quadrupole mass spectrometer, Waters Corporation (Maple street Milford, MA, USA) equipped with an electrospray ionization (ESI) source; chromatography was carried on an Acquity H Class Plus UPLC system; quaternary solvent manager, sample manager FTN-H and column oven (New York, USA).

Chromatographic separation was achieved using an Acquity C18 -BEH column (1.7 µm, 2.1 × 50 mm), Waters, USA. Isocratic elution was achieved using the binary mobile phase consisting of 0.01 M ammonium formate pH 8.0: acetonitrile (47 : 53, v/v) as a mobile phase pumped at a flow rate of 0.2 ml min$^{-1}$. The MS detection method was carried out in the positive ESI mode. The quadrupole mass spectrometer operated at the multiple reaction monitoring (MRM) mode, monitoring the transition of molecular ions to the product ions in positive ion mode at m/z 298.16/96.89 for VBR, m/z 384.11/ 67.96 for MRP and m/z 555.09/ 396.03. For internal standard (IS), the operating conditions were optimized to monitor the three compounds as summarized in table 1. The cone gas was nitrogen, whereas argon was used as the collision gas. The source/gas-dependent parameters were as

follows: dessolvation gas flow 1000 (l h$^{-1}$); cone gas flow, 50 (l h$^{-1}$); dessolvation temperature, 600°C and Capillary voltage, 4 (kV).

## 2.4. Preparation of calibration standard

Ceftriaxone (IS) of concentration 150.00 µg ml$^{-1}$ was prepared by accurately weighing 15.00 mg of ceftriaxone into a 100 ml volumetric flask, 60.0 ml methanol were added and the volume was completed with the same solvent. Stock solutions of concentration 100.00 µg ml$^{-1}$ VBR and MRP for calibration were prepared separately by accurately weighing 10.00 mg of each drug and transferred into 100 ml volumetric flasks, 60.0 ml of methanol was added to each flask, then the volume was completed with the same solvent. A series of calibrator working solutions of VBR and MRP at concentration levels of 5.00, 20.00, 30.00, 50.00, 100.00, 300.00, 450.00 and 500.00 µg ml$^{-1}$ were prepared from their stock solutions applying a serial dilution technique using methanol as the diluting solvent. Also, working quality control samples solutions were prepared by the same technique to prepare three levels of concentration 15.00, 150.00 and 400.00 µg ml$^{-1}$. All the prepared solutions were kept in the refrigerator at 2–8°C.

## 2.5. Construction of calibration curves

Calibration standards of VBR and MRP in plasma were prepared at concentration levels of 0.50, 2.00, 3.00, 5.00, 10.00, 30.00, 45.00, and 50.00 µg ml$^{-1}$ by spiking 50.0 µl of each calibrator working solution into 400.0 µl blank plasma. The quality control samples (QCs) at three concentration levels; low (QCL, 1.50 µg ml$^{-1}$), medium (QCM, 15.00 µg ml$^{-1}$), and high (QCH, 40.00 µg ml$^{-1}$) were prepared from their working QC samples solutions as described for calibration standards. The spiked plasma, calibrators and QC samples are then treated as described below under sample preparation.

## 2.6. Plasma sample preparation

A volume of 50.0 µl aliquots of IS solution, 150.00 µg ml$^{-1}$, was added to 500.0 µl of each spiked plasma sample and mixed with the aid of vortex for 10 s. Then, 3.0 ml of acetonitrile was mixed and vortexed for 4 min. The solution was centrifuged at 5000 rpm for 5 min. Finally, a volume of 1.0 µl of supernatant was injected into the UPLC-MS/MS system. The study was carried out in triplicate.

## 2.7. Application to pharmaceutical preparations

An accurate amount equivalent to 100 mg of each of VBR and MRP from Vabomere® powder for injection was weighed, carefully transferred and dissolved in acetonitrilein 100 ml volumetric flask, then further dilution has been performed in methanol to prepare a solution containing 100 µg ml$^{-1}$ of each drug. The procedure was continued, as stated under §§2.5 & 2.6. The per cent recoveries were calculated by referring to the calibration graphs or using the corresponding intra-run regression equations.

## 2.8. Method validation

Validation of the developed technique was accomplished to achieve best practices for the analytical and bioanalytical method validation [19,20].

### 2.8.1. Selectivity and specificity

UPLC, coupled with MS detector with the MRM mode, has the advantage of being more selective than other non-hyphenated techniques. Recognition of any extra peaks was performed by monitoring the chromatograms of the samples. Method selectivity was tested by analysing six different sets of plasma samples with UPLC applying the MRM mode.

### 2.8.2. Linearity

It was assessed by plotting the peak area ratio of the MS transition pair of analytes to that of IS against the nominated concentration of standard calibration solution. The concentration ranges used ranged from

$0.50$ to $50.00\ \mu g\ ml^{-1}$, for both VBR and MRP. The two calibration curves were repeated and constructed six times for each drug in plasma. Samples correspond to blank human plasma, and QCs were analysed parallel to each calibration curve.

### 2.8.3. Precision and accuracy

Inter- and intra-assay precision and accuracy were evaluated by investigating six repeats at the lower level of quantification (LLOQ) in addition to three different QC levels (QCL, QCM and QCH) within the same and different days.

### 2.8.4. Recovery

The recovery of VBR and MRP from plasma was assessed by quantifying the percentage of the responses of the analytes obtained from replicate QC samples with the response of analytes from post extracted plasma standard samples at the same concentrations at the three levels.

### 2.8.5. Matrix effect

To assess the consequence of plasma components on the ionization of analytes and IS, the responses of the post extracted plasma standard at the three QC levels ($n = 4$) were compared with their response from neat samples.

### 2.8.6. Dilution integrity

Accuracy of samples, if needed to be diluted to lie within the linearity range, was assessed. Dilution accuracy was studied to confirm that the dilution procedure has no impact on the measured concentration. VBR and MRP were spiked to human plasma samples and diluted with pooled human plasma two- and fourfold in six repeats and then investigated. The six repeats should have a precision of $\leq 15\%$ and an accuracy of $100 \pm 15\%$.

## 2.9. Stability experiments

The effect of the storage conditions of analytes and IS was determined periodically by injecting replicate preparations of handled samples up to 12 h at room temperature. The peak areas of the analytes and IS achieved at the initial time were used as a baseline value to determine the relative stability of the analytes at the following points. It was determined at two concentration levels in six repeats. Freeze stability of the analytes in plasma was evaluated by studying the QCL and QCH samples stored at $-80 \pm 10°C$ for at least 90 days. The stability of analytes in plasma following repeated three freeze–thaw cycles were evaluated.

# 3. Results and discussion

In 2017, the FDA approved Vabomere® for adults with complicated urinary tract infections, including a type of kidney infection, pyelonephritis, caused by specific bacteria. Vabomere® is a drug containing MRP, an antibacterial drug, and VBR, which is added to inhibit certain types of resistance mechanisms used by bacteria [21]. To date, to our knowledge, there is no technique that has been published for the evaluation of both drugs in human plasma, and with the expected upcoming generics production, there is a need to establish a bioanalytical method that could be applied as a requirement for registration of such generics and prove their bioequivalence to brand new, Vabomere®. Therefore, the main aim of this study was to launch and validate a UPLC-MS/MS method for the concurrent determination of VBR and MRP in the concentration range around their $C_{max}$ ($28\ mg\ l^{-1}$ for VBR and $30\ mg\ l^{-1}$ for MRP) attained in human plasma following 1 g dose administration of each drug. For optimum detection of the studied compounds in parallel with the IS, both the mass spectrometric parameters and the chromatographic conditions were studied.

## 3.1. Optimization of sample preparation for spiked plasma

Sample preparation is a critical step for drug determination in human plasma. Various ways were tried as liquid–liquid extraction (using ethyl acetate, diethyl ether, dichloromethane and n-hexane) and

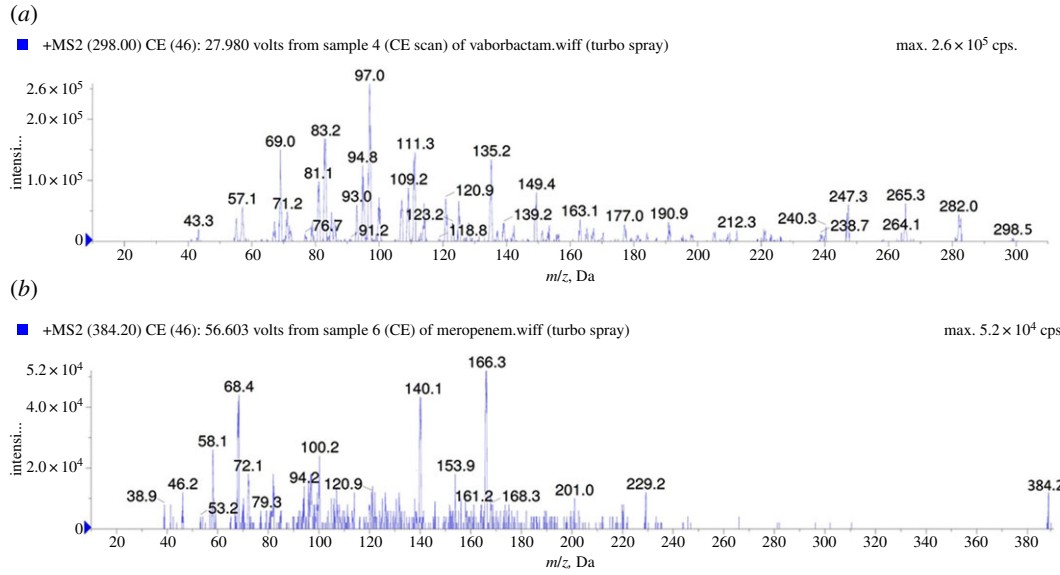

**Figure 2.** MS/MS spectra of [M + H]$^+$ of vaborbactam of $m/z$ 298.0 (*a*) and meropenem of $m/z$ 384.2 (*b*).

precipitation techniques (using methanol and acetonitrile) for simultaneous determination of VBR and MRP. The highest recoveries were obtained from human plasma upon using acetonitrile as a precipitating solvent. Then 1.0 µl was injected into the UPLC–MS/MS system.

## 3.2. Optimization of chromatographic and mass spectrometric conditions

According to the molecular structure of the two studied compounds, they form positive molar-ions under acidic chromatographic conditions. ESI in the positive mode was used for MRM analysis. The MRM spectra along with IS showing the precursor produced at (Q1) and selected product ion (Q3) is presented in figure 2. As shown in figure 2, the most abundant product ion at 96.89 $m/z$ for VBR was selected, while for MRP that corresponding to 166 $m/z$ was first selected but low reproducibly and selectivity was observed, so the product ion at 67.96 was selected and proved to be reproducible and showed good sensitivity. The Q1 full-scan mass spectra of VBR, MRP and IS showed predominant protonated precursor [M + H]$^+$ ions at $m/z$ 298.16, 384.11 and 555.09, respectively. Detection of ions was performed in the MRM mode by monitoring the transition pairs as described under the experimental section. The suggested product ion (Q3) for each of the studied compounds is shown in figure 1.

In order to optimize the proposed UPLC–MS/MS method, the effects of several chromatographic parameters were investigated. Different columns were used; Acquity (C8-BEH, (1.7 µm, 2.1 × 50 mm), Kinetex® C8 (2.6 µm, 100 Å, 50 × 2.1 mm) and the optimum column that was used, Acquity C18-BEH (1.7 µm, 2.1 × 50 mm) that gave a high resolution. For optimization of the mobile phase composition, different parameters were tested and optimized as the type of aqueous phase and organic modifier, organic modifier-aqueous ratio and pH. Different aqueous solutions were tried in different percentages, acetonitrile (from 40 to 50%), 0.1% formic (from 30 to 50%) and 0.1% acetic acid (from 30 to 50%) 0.01 M ammonium acetate (from 30 to 50%), 0.01 and 0.015 M ammonium formate (from 40 to 50%). For the organic phase, acetonitrile (from 40 to 60%) was tried. Acetic acid, formic acid and ammonium hydroxide were used to adjust the pH range from 5.5 up to 8.0. These parameters were optimized based on the peak shape, peak intensity/area, peak resolution and retention time for the analytes on the Acquity C18-BEH (1.7 µm, 2.1 × 50 mm) column. It was observed that the composition and pH of the mobile phase had a significant impact on separation selectivity and sensitivity of the method. The sensitivity was significantly increased with the use of 0.01 M ammonium formate pH 8.0: acetonitrile (47 : 53, v/v) as a mobile phase with a flow rate of 0.2 ml min$^{-1}$. The samples were run at the optimized chromatographic conditions under the general MS parameters and the specified parameters for each compound, table 1. The small-volume injection has an advantage in prolonging the life duration of smaller columns, saving of device time and minor eluent consumption together with a cleaner mass source. All the analytes and IS were eluted in the narrow range of retention times (0.3–0.4 min), which is advantageous for the compensation of matrix effects. Extracted multiple

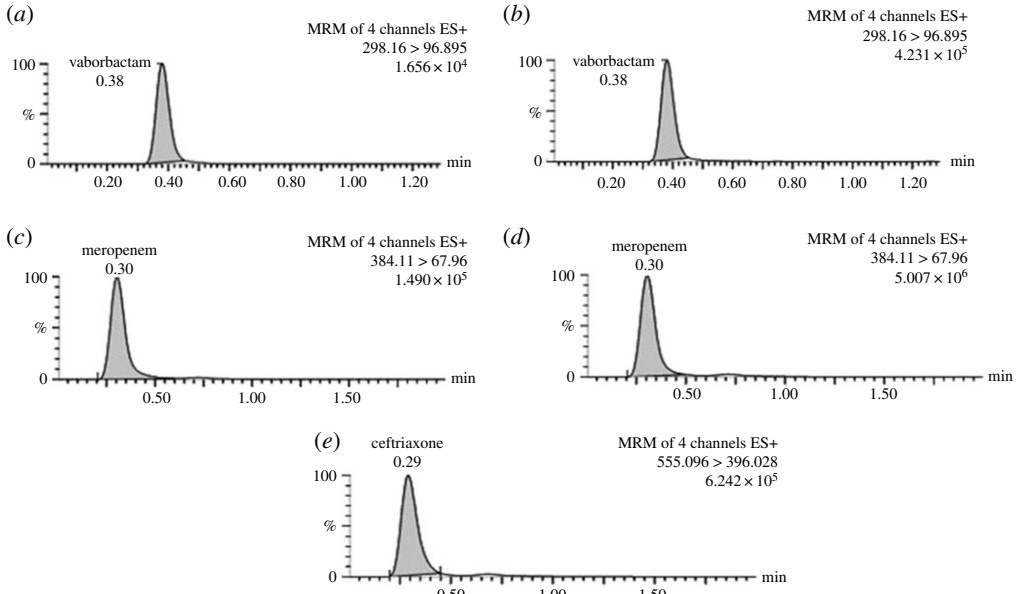

**Figure 3.** Extracted multiple reaction monitoring chromatograms of plasma samples spiked with (*a*) vaborbactam at QCL level (1.50 µg ml$^{-1}$); (*b*) vaborbactam at QCH level (40.00 µg ml$^{-1}$); (*c*) meropenem at QCL level (1.50 µg ml$^{-1}$); (*d*) meropenem at QCH level (40.00 µg ml$^{-1}$) and (*e*) IS.

reaction monitoring chromatograms of spiked plasma samples with VBR and MRP at QCL level (1.50 µg ml$^{-1}$) and QCH level (40.00 µg ml$^{-1}$) and IS are illustrated in figure 3.

## 3.3. Method validation

### 3.3.1. Selectivity and specificity

The chromatograms of the samples were monitored for the detection of any extra peaks. However, no chromatographic interference from any of the excipients was observed at the retention times of the drugs. These results revealed the absence of interference from any ingredients in the dosage forms. Moreover, the effect of the co-extracted biogenic materials from plasma was studied, and the chromatograms of blank human plasma and blank plasma spiked with IS and analytes revealed that the minimal effect of these endogenous materials was at retention times 0.30 and 0.38 min for MRP and VBR, respectively, which confirms the selectivity of the proposed method. Also, the specificity of the proposed method was evaluated via analysis of six different batches of control human plasma and the chromatographic interference was evaluated statistically by calculating the per cent relative area of interfering peak to area of the studied drugs and IS at its LLOQ, and all of these relative ratios were less than 20% which revealed the efficient specificity of the proposed method, this is presented in the electronic supplementary material table S1.

### 3.3.2. Linearity

Human free drug plasma was used as a matrix for calibration. Under the optimized experimental conditions, linearity was evaluated by analysing eight different concentrations of each compound in triplicate. A linear relationship between the peak area ratios of each analyte/IS, and their corresponding concentrations were established. The results revealed that the peak area ratios of VAR and MRP to IS were found to be linear over the concentration range, 0.5 to 50 µg ml$^{-1}$ for both drugs. The obtained slope, intercept and correlation coefficient values for each compound in plasma were calculated. The results revealed the dynamic linearity of the proposed method as the regression coefficients were 0.9962 with slope value 0.0706 ± 0.0012 and 0.9956 with slope value 0.1624 ± 0.0020 for VAR and MRP, respectively. The linear curves were accurate as the intercepts were very small and equal to −0.0078 and −0.0255 for VAR and MRP, respectively. The LLOQ for each compound in plasma was calculated and found to be 0.50 µg ml$^{-1}$ for both drugs.

**Table 2.** Intra-assay and inter-assay precision and accuracy of vaborbactam and meropenem in plasma.

| component | concentration (µg ml$^{-1}$) | | bias (%)[a] | CV (%)[a] |
|---|---|---|---|---|
| | added | found[a] | | |
| intra-assay precision and accuracy | | | | |
| vaborbactam | LLOQ (0.50) | 0.48 | −3.30 | 5.13 |
| | QCL (1.50) | 1.31 | −12.92 | 1.67 |
| | QCM (15.00) | 14.00 | −6.64 | 4.44 |
| | QCH (40.00) | 38.14 | −4.65 | 4.12 |
| meropenem | LLOQ (0.50) | 0.54 | 7.23 | 3.24 |
| | QCL (1.50) | 1.31 | −12.95 | 1.04 |
| | QCM (15.00) | 13.17 | −12.21 | 1.42 |
| | QCH (40.00) | 36.92 | −3.08 | 4.58 |
| inter-assay precision and accuracy | | | | |
| vaborbactam | LLOQ (0.50) | 0.52 | 2.00 | 10.54 |
| | QCL (1.50) | 1.32 | −2.00 | 3.24 |
| | QCM (15.00) | 13.43 | −3.80 | 5.50 |
| | QCH (40.00) | 40.90 | 2.25 | 6.31 |
| meropenem | LLOQ (0.50) | 0.55 | 10.00 | 7.41 |
| | QCL (1.50) | 1.30 | −13.33 | 1.51 |
| | QCM (15.00) | 13.80 | −8.00 | 4.67 |
| | QCH (40.00) | 39.32 | −1.70 | 5.81 |

[a]The calculated values are the average of three determinations of each sample during intra- and inter-daily analysis.

### 3.3.3. Precision and accuracy

Both inter- and intra-day precision and accuracy were calculated at four levels for each of the studied drugs, LLOQ, QCL, QCM and QCH. The precision was evaluated statistically by coefficient of variation (CV) of the found amount as a percentage, while the relative percentage error (bias %) was calculated to evaluate the accuracy. The results showed good agreement with the guidelines acceptance criteria, as shown in table 2. The intra-assay precision, as % CV values, were within a range of 1.67–5.13% and 1.04–4.58 with calculated bias % in the range of 3.30–12.92 and 3.08–12.95 for VAR and MRP, respectively. For inter-assay precision, the % CV was within a range of 3.24–10.54 and 1.51 to 7.41 with calculated bias % in the range of 2.00–3.80 and 1.70–13.33 for VAR and MRP, respectively. The obtained results revealed that both the % CV and % bias does not exceed 15% for both drugs, which comply with the FDA bioanalytical guidelines [19].

### 3.3.4. Extraction recovery

The analysed compounds were efficiently recovered from spiked plasma, calculated at three QC levels and found to be 77.18, 79.94 and 78.93% for VBR and 77.66, 80.09 and 79.00% for MRP, respectively. The mean extraction recovery for IS was calculated and found to be 82.24%.

### 3.3.5. Matrix effects

The effect of plasma constituents over the ionization of analytes and IS was determined by comparing the responses of the post extracted plasma standard QC samples ($n = 4$) with the response of analytes from neat samples at equivalent concentrations. The relative standard deviation of peak area ratios (analyte/IS) was lower than 2%, and the relative standard deviation of peak areas of individual compounds was lower than 4%, indicating no significant matrix effects.

**Table 3.** Stability of vaborbactam and meropenem in the matrix by the proposed method.

| parameter | stability %[a] | |
| --- | --- | --- |
| | vaborbactam | meropenem |
| short-term stability of analyte in the matrix at room temperature (12 h) | | |
| QCL | 101.44 | 98.76 |
| QCH | 99.79 | 99.08 |
| long-term stability of analyte in matrix at −80 °C (70 days) | | |
| QCL | 102.41 | 101.70 |
| QCH | 98.42 | 101.16 |
| freeze and thaw stability | | |
| QCL | | |
| cycle 1 | 95.81 | 99.33 |
| cycle 2 | 105.01 | 100.16 |
| cycle 3 | 99.97 | 103.26 |
| QCH | | |
| cycle 1 | 100.75 | 101.95 |
| cycle 2 | 99.11 | 94.54 |
| cycle 3 | 101.52 | 101.04 |

[a]Each value was calculated as an average of three replicate analysis.

### 3.3.6. Sample stability

Bioanalytical method validation requires the study of the sample stability under different conditions. All stability parameters were tested for spiked plasma samples, and the results are shown in table 3. The short-term stability of analytes in plasma samples (with low- and high-quality control samples) was studied for period of 12 h at room temperature (25°C) and ambient light, while the long-term stability of frozen plasma samples was examined after 70 days storage at −80°C. The samples were stable under the studied conditions and the average percentage recoveries were not less than 98% for both drugs at different concentration levels. For freeze and thaw stability, plasma samples with low and high concentrations of the two analytes were prepared. The samples were stored at −20°C and subjected for three freeze–thaw cycles. During each cycle triplicate, 1 ml aliquot was processed and analysed, and the results showed non-significant substance loss during repeated thawing and freezing cycles, as shown in table 3.

### 3.3.7. Application to pharmaceutical preparations

The proposed method was effectively applied for the analysis of the studied compounds in pharmaceutical formulation in order to further assess the selectivity of the proposed UPLC-MS/MS in the dosage form matrix. The concentrations of the drugs were calculated, referring to the corresponding regression equation. The mean percentage recoveries obtained for the two drugs were $97.05 \pm 1.02$ and $100.34 \pm 1.98\%$ for vaborbactam and meropenem, respectively.

## 4. Conclusion

A novel, sensitive, simple and selective UPLC-MS/MS method was developed for the simultaneous determination of VBR and MRP in pharmaceutical dosage form and human plasma. The use of UPLC has enabled the use of a short column with a small particle size, which has resulted in sharp chromatographic peaks within a short run time. Thus, it is possible to analyse a large number of plasma samples per day. Besides, the low flow rate applied in this method has offered less solvent consumption, which is considered to be cost-effective and eco-friendly. The proposed method has proved to have suitable LLOQ of 0.5 µg ml$^{-1}$, which is far away from their expected $C_{max}$ for each of VBR and MRP, which permits the use of the method for pharmacokinetic studies. The method was validated in accordance with FDA and

International Conference on Harmonization guidelines. The results gained from the validation study have confirmed that the new method is selective, linear, precise and accurate. Additionally, the developed assay method could be successfully applied to the bioequivalence evaluation of generic drugs.

Ethics. Human plasma samples were collected and handled using the ethics protocol reviewed and approved by the bioethical committee of King Abdulaziz University, under grant no. (G-248–166-1440).

Data accessibility. The data supporting the results in this article can be accessed from the Dryad Digital Repository: https://doi.org/10.5061/dryad.866t1g1nc [22].

Authors' contributions. A.K.K. and A.K. have been involved in designing, acquisition of data, and interpretation of data. A.N.K. was involved in designing and revising of the manuscript. A.K.K. and M.A.H. did experimental work supervision, analysis of the data, conducting method validation and drafting the article. All authors revised and approved the final form of the manuscript.

Competing interests. We declare we have no competing interests.

Funding. This project was funded by the Deanship of Scientific Research (DSR), King Abdulaziz University, Jeddah, under grant no. (G-248-166-1440).

Acknowledgements. The authors, therefore, gratefully acknowledge the DSR technical and financial support.

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
