## [Reviewer comments · Royal Society Open Science]

Review History

RSOS-200635.R0 (Original submission)

Review form: Reviewer 1 (Mohamed Attwa)

Is the manuscript scientifically sound in its present form?

Yes

Are the interpretations and conclusions justified by the results?

Yes

Is the language acceptable?

Yes

Do you have any ethical concerns with this paper?

No

Have you any concerns about statistical analyses in this paper?

No

Recommendation?

Accept with minor revision (please list in comments)

Comments to the Author(s)

Following my review, some points have to be considered before going through acceptance and publication.

- 1) Results and discussion part, sec 3: The product ion, m/z 97, was selected as the most abundant MS₂ ion of VBR, this is logic. However, m/z 68.4 was selected as product ion for MRM monitoring of MRP. Why did authors not select m/z 166.3?
- 3) Page 3, line 7, sec 2.1-, is the purity stated a supplier certified or based on analysis done by the authors?
- 3) Page 5, sec. 2.7, acetonitrile is used for dilution, which is different than the solvent used in sec 2.4, clarify? Justify.
- 4) Page 5, sec 2.8.3: Check the significant figures in standards concentration values.
- 5) Reference no.14: has to be changed to the updated version of bioanalytical validation.
Author should cite the following reference to approve the capability of LC-MS/MS technique to easily quantify and detect drugs in different biological matrices:
 - a- Kadi, A. A., Abdelhameed, A. S., Darwish, H. W., Attwa, M. W., & Bakheit, A. H. (2016). Liquid chromatographic-tandem mass spectrometric assay for simultaneous quantitation of tofacitinib, cabozantinib and afatinib in human plasma and urine. *Tropical Journal of Pharmaceutical Research*, 15(12), 2683-2692.
 - b- Amer, S. M., Kadi, A. A., Darwish, H. W., & Attwa, M. W. (2017). Liquid chromatography tandem mass spectrometry method for the quantification of vandetanib in human plasma and rat liver microsomes matrices: metabolic stability investigation. *Chemistry Central Journal*, 11(1), 45.
 - c- Attwa, M. W., Kadi, A. A., Darwish, H. W., Amer, S. M., & Alrabiah, H. (2018). A reliable and stable method for the determination of foretinib in human plasma by LC-MS/MS: application to metabolic stability investigation and excretion rate. *European Journal of Mass Spectrometry*, 24(4), 344-351.
 - d- Attwa, M. W., Kadi, A. A., Darwish, H. W., & Abdelhameed, A. S. (2018). Investigation of the metabolic stability of olmutinib by validated LC-MS/MS: quantification in human plasma. *RSC advances*, 8(70), 40387-40394.
- 6) Page 5, sec 2.8.1- and page 9, sec 3.3.1: deal with the same validation parameter with different identification, they shall be the same, Justify.
- 7) Page 8, line 48, the column name must be matched with that mentioned in abstract line 58, Acquity C18-BEH.
- 8) Page, table 2- has to be moved to supplementary data.
- 9) Page 11, sec 3.3.3- bioanalytical method validation is not cited.
- 10) Page 12, line 20 refers to within-day analysis while the table includes both intra- and inter-day assay precision.
- 11) Page 13, table 4- is the calculated values for each sample were obtained from single or replicate analysis?
- 12) The Manuscript text needs careful revision against English and grammar, as well as fragmenting the long sentences.

Review form: Reviewer 2

Is the manuscript scientifically sound in its present form?

Yes

Are the interpretations and conclusions justified by the results?

Yes

Is the language acceptable?

Yes

Do you have any ethical concerns with this paper?

No

Have you any concerns about statistical analyses in this paper?

No

Recommendation?

Accept with minor revision (please list in comments)

Comments to the Author(s)

The authors described a new method to determine the compounds of VBR and MRP in human plasma using UPLC-MS/MS with good selectivity, linearity, precision and accuracy. The reported study was well executed and will find potential applications in clinical use. Thus it is suggested to be published. Here are some comments for the authors to consider:

1. The author should provide the ethical approvals about the human plasma in this research.
2. In references, the format is not unified. Please check them one by one carefully.

Review form: Reviewer 3

Is the manuscript scientifically sound in its present form?

Yes

Are the interpretations and conclusions justified by the results?

Yes

Is the language acceptable?

Yes

Do you have any ethical concerns with this paper?

No

Have you any concerns about statistical analyses in this paper?

No

Recommendation?

Accept as is

Comments to the Author(s)

The authors have done sufficient work to support their conclusion.

I have some reservations about the suitability of the method study, but I choose to trust the academic judgment they gave and show respect to the academic and professional accomplishment of all the authors.

Review form: Reviewer 4

Is the manuscript scientifically sound in its present form?

Yes

Are the interpretations and conclusions justified by the results?

Yes

Is the language acceptable?

Yes

Do you have any ethical concerns with this paper?

No

Have you any concerns about statistical analyses in this paper?

No

Recommendation?

Reject

Comments to the Author(s)

Dear editor,

The manuscript showed a novel ultra-performance liquid chromatography (UPLC)- tandem mass spectrometric (MS/MS) method was developed for the sensitive determination of both compounds in human plasma. The manuscript was well-written and the results were credible. However, a article has been published in Analytical and Bioanalytical Chemistry with the title of A validated LC-MSMS method for the simultaneous quantification of meropenem and vaborbactam in human plasma and renal replacement therapy effluent and its application to a pharmacokinetic study (DOI: 10.1007/s00216-019-02184-4). The LLOQ in that article was 0.05 μ g mL⁻¹. Therefore the paper cannot be accepted for publication in Royal Society Open Science.

Decision letter (RSOS-200635.R0)

Dear Dr Hegazy:

Title: UPLC-MS/MS Method for quantitation of the recently FDA approved combination of vaborbactam and meropenem in human plasma

Manuscript ID: RSOS-200635

The editor assigned to your manuscript has now received comments from reviewers. We would like you to revise your paper in accordance with the referee and Subject Editor suggestions which can be found below (not including confidential reports to the Editor). Please note this decision does not guarantee eventual acceptance.

Please submit your revised paper before 27-Jun-2020. Please note that the revision deadline will expire at 00.00am on this date. If we do not hear from you within this time then it will be assumed that the paper has been withdrawn. In exceptional circumstances, extensions may be possible if agreed with the Editorial Office in advance. We do not allow multiple rounds of revision so we urge you to make every effort to fully address all of the comments at this stage. If deemed necessary by the Editors, your manuscript will be sent back to one or more of the original reviewers for assessment. If the original reviewers are not available we may invite new reviewers.

Please also include the following statements alongside the other end statements. As we cannot publish your manuscript without these end statements included, if you feel that a given heading is not relevant to your paper, please nevertheless include the heading and explicitly state that it is not relevant to your work.

- Ethics statement

Please clarify whether you received ethical approval from a local ethics committee to carry out your study. If so please include details of this, including the name of the committee that gave consent in a Research Ethics section after your main text. Please also clarify whether you received informed consent for the participants to participate in the study and state this in your Research Ethics section.

OR

Please clarify whether you obtained the necessary licences and approvals from your institutional animal ethics committee before conducting your research. Please provide details of these licences and approvals in an Animal Ethics section after your main text.

OR

Please clarify whether you obtained the appropriate permissions and licences to conduct the fieldwork detailed in your study. Please provide details of these in your methods section.

Royal Society of Chemistry
Thomas Graham House
Science Park, Milton Road
Cambridge, CB4 0WF

Royal Society Open Science - Chemistry Editorial Office

RSC Associate Editor:
Comments to the Author:
(There are no comments.)

RSC Subject Editor:
Comments to the Author:
(There are no comments.)

Reviewers' Comments to Author:
Reviewer: 1

Comments to the Author(s)
Following my review, some points have to be considered before going through acceptance and publication.

- 1) Results and discussion part, sec 3: The product ion, m/z 97, was selected as the most abundant MS₂ ion of VBR, this is logic. However, m/z 68.4 was selected as product ion for MRM monitoring of MRP. Why did authors not select m/z 166.3?
- 3) Page 3, line 7, sec 2.1-, is the purity stated a supplier certified or based on analysis done by the authors?
- 3) Page 5, sec. 2.7, acetonitrile is used for dilution, which is different than the solvent used in sec 2.4, clarify? Justify.
- 4) Page 5, sec 2.8.3: Check the significant figures in standards concentration values.
- 5) Reference no.14: has to be changed to the updated version of bioanalytical validation. Author should cite the following reference to approve the capability of LC-MS/MS technique to easily quantify and detect drugs in different biological matrices:
 - a- Kadi, A. A., Abdelhameed, A. S., Darwish, H. W., Attwa, M. W., & Bakheit, A. H. (2016). Liquid chromatographic-tandem mass spectrometric assay for simultaneous quantitation of tofacitinib, cabozantinib and afatinib in human plasma and urine. *Tropical Journal of Pharmaceutical Research*, 15(12), 2683-2692.
 - b- Amer, S. M., Kadi, A. A., Darwish, H. W., & Attwa, M. W. (2017). Liquid chromatography tandem mass spectrometry method for the quantification of vandetanib in human plasma and rat liver microsomes matrices: metabolic stability investigation. *Chemistry Central Journal*, 11(1), 45.
 - c- Attwa, M. W., Kadi, A. A., Darwish, H. W., Amer, S. M., & Alrabiah, H. (2018). A reliable and stable method for the determination of foretinib in human plasma by LC-MS/MS: application to metabolic stability investigation and excretion rate. *European Journal of Mass Spectrometry*, 24(4), 344-351.
 - d- Attwa, M. W., Kadi, A. A., Darwish, H. W., & Abdelhameed, A. S. (2018). Investigation of the metabolic stability of olmutinib by validated LC-MS/MS: quantification in human plasma. *RSC advances*, 8(70), 40387-40394.

- 6) Page 5, sec 2.8.1- and page 9, sec 3.3.1: deal with the same validation parameter with different identification, they shall be the same, Justify.
- 7) Page 8, line 48, the column name must be matched with that mentioned in abstract line 58, Acquity C18-BEH.
- 8) Page, table 2- has to be moved to supplementary data.
- 9) Page 11, sec 3.3.3- bioanalytical method validation is not cited.
- 10) Page 12, line 20 refers to within-day analysis while the table includes both intra- and inter-day assay precision.
- 11) Page 13, table 4- is the calculated values for each sample were obtained from single or replicate analysis?
- 12) The Manuscript text needs careful revision against English and grammar, as well as fragmenting the long sentences.

Reviewer: 2

Comments to the Author(s)

The authors described a new method to determine the compounds of VBR and MRP in human plasma using UPLC-MS/MS with good selectivity, linearity, precision and accuracy. The reported study was well executed and will find potential applications in clinical use. Thus it is suggested to be published. Here are some comments for the authors to consider:

1. The author should provide the ethical approvals about the human plasma in this research.
2. In references, the format is not unified. Please check them one by one carefully.

Reviewer: 3

Comments to the Author(s)

The authors have done sufficient work to support their conclusion.

I have some reservations about the suitability of the method study, but I choose to trust the academic judgment they gave and show respect to the academic and professional accomplishment of all the authors.

Reviewer: 4

Comments to the Author(s)

Dear editor,

The manuscript showed a novel ultra-performance liquid chromatography (UPLC)- tandem mass spectrometric (MS/MS) method was developed for the sensitive determination of both compounds in human plasma. The manuscript was well-written and the results were credible. However, a article has been published in *Analytical and Bioanalytical Chemistry* with the title of A validated LC-MSMS method for the simultaneous quantification of meropenem and vaborbactam in human plasma and renal replacement therapy effluent and its application to a pharmacokinetic study (DOI: 10.1007/s00216-019-02184-4). The LLOQ in that article was 0.05 μ g mL⁻¹. Therefore the paper cannot be accepted for publication in Royal Society Open Science.

Author's Response to Decision Letter for (RSOS-200635.R0)

See Appendix A.

Decision letter (RSOS-200635.R1)

Dear Dr Hegazy:

Title: UPLC-MS/MS Method for quantitation of the recently FDA approved combination of vaborbactam and meropenem in human plasma
Manuscript ID: RSOS-200635.R1

It is a pleasure to accept your manuscript in its current form for publication in Royal Society Open Science. The chemistry content of Royal Society Open Science is published in collaboration with the Royal Society of Chemistry.

RSC Associate Editor
Comments to the Author:
(There are no comments.)

Reviewer(s)' Comments to Author:

Appendix A

Dear Editor;

I hope everything is going well in the circumstances of COVID 19. This response is regarding the manuscript that have been honorary submitted to your reputable journal. The article is under the title of : UPLC-MS/MS Method for quantitation of the recently FDA approved combination of vaborbactam and meropenem in human plasma. And assigned the following no. Manuscript ID: RSOS-200635

all reviewers' comments have been considered and taken into account according to the followings:

Comment	Response
Reviewer: 1	
Results and discussion part, sec 3: The product ion, m/z 97, was selected as the most abundant MS2 ion of VBR, this is logic. However, m/z 68.4 was selected as product ion for MRM monitoring of MRP. Why did authors not select m/z 166.3?	Added to results and discussion, and highlighted.
Page 3, line 7, sec 2.1-, is the purity stated a supplier certified or based on analysis done by the authors?	The purity was based on the COA, adjusted and highlighted.
Page 5, sec. 2.7, acetonitrile is used for dilution, which is different than the solvent used in sec 2.4, clarify? Justify.	Acetonitrile was the best solvent used during determination for the stated drugs in pharmaceutical formulation, then all other dilutions are performed by methanol as stated under 2.7, that the procedure under 2.5 and 2.6 was followed after preparation of the first stock solution of dosage form in acetonitrile. The statement has been corrected to be clearer and highlighted.
Page 5, sec 2.8.3: Check the significant figures in standards concentration values.	All significant figures have been reviewed and adjusted, highlighted as well.
Reference no.14: has to be changed to the updated version of bioanalytical validation.	Reference no. 14 has been updated to the latest version (May 2018 instead of 2001 and highlighted), and now after modification of manuscript updated to be no. 19
Author should cite the following reference to approve the capability of LC-MS/MS technique to easily quantify and detect drugs in different biological matrices:	
a- Kadi, A. A., Abdelhameed, A. S., Darwish, H. W., Attwa, M. W., & Bakheit, A. H. (2016). Liquid chromatographic-tandem mass spectrometric assay for simultaneous quantitation of tofacitinib, cabozantinib and afatinib in human plasma and urine. Tropical	Added to the manuscript and highlighted

Journal of Pharmaceutical Research, 15(12), 2683-2692.	
b- Amer, S. M., Kadi, A. A., Darwish, H. W., & Attwa, M. W. (2017). Liquid chromatography tandem mass spectrometry method for the quantification of vandetanib in human plasma and rat liver microsomes matrices: metabolic stability investigation. Chemistry Central Journal, 11(1), 45.	Added to the manuscript and highlighted
c- Attwa, M. W., Kadi, A. A., Darwish, H. W., Amer, S. M., & Alrabiah, H. (2018). A reliable and stable method for the determination of foretinib in human plasma by LC-MS/MS: application to metabolic stability investigation and excretion rate. European Journal of Mass Spectrometry, 24(4), 344-351.	Added to the manuscript and highlighted
d- Attwa, M. W., Kadi, A. A., Darwish, H. W., & Abdelhameed, A. S. (2018). Investigation of the metabolic stability of olmutinib by validated LC-MS/MS: quantification in human plasma. RSC advances, 8(70), 40387-40394.	Added to the manuscript and highlighted
Page 5, sec 2.8.1- and page 9, sec 3.3.1: deal with the same validation parameter with different identification, they shall be the same, Justify.	Corrected under 2.8.1 as per the bioanalytical guidelines and highlighted.
Page 8, line 48, the column name must be matched with that mentioned in abstract line 58, Acquity C18-BEH.	Corrected and now the column name is matched along the manuscript.
Page, table 2- has to be moved to supplementary data.	The table has been added to supplementary data (Table S1).
Page 11, sec 3.3.3- bioanalytical method validation is not cited.	Citation adjusted and highlighted.
Page 12, line 20 refers to within-day analysis while the table includes both intra- and inter-day assay precision.	Corrected and highlighted.
Page 13, table 4- is the calculated values for each sample were obtained from single or replicate analysis?	A footnote is added showing the replicate analysis used and highlighted.
The Manuscript text needs careful revision against English and grammar, as well as fragmenting the long sentences.	The text has been revised and long sentences have been fragmented as needed for better understanding.
Reviewer: 2	
The authors described a new method to determine the compounds of VBR and MRP	Your effort and time in reviewing the manuscript are highly appreciated.

in human plasma using UPLC-MS/MS with good selectivity, linearity, precision and accuracy. The reported study was well executed and will find potential applications in clinical use. Thus it is suggested to be published. Here are some comments for the authors to consider:	Thank you so much for your comment.
The author should provide the ethical approvals about the human plasma in this research.	Added and highlighted
In references, the format is not unified. Please check them one by one carefully.	Checked and unified
Reviewer: 3	
The authors have done sufficient work to support their conclusion. I have some reservations about the suitability of the method study, but I choose to trust the academic judgment they gave and show respect to the academic and professional accomplishment of all the authors.	Your effort and time in reviewing the manuscript are highly appreciated. Thank you so much for your comment.
Reviewer: 4	
The manuscript showed a novel ultra-performance liquid chromatography (UPLC)-tandem mass spectrometric (MS/MS) method was developed for the sensitive determination of both compounds in human plasma. The manuscript was well-written and the results were credible. However, a article has been published in Analytical and Bioanalytical Chemistry with the title of A validated LC-MSMS method for the simultaneous quantification of meropenem and vaborbactam in human plasma and renal replacement therapy effluent and its application to a pharmacokinetic study (DOI: 10.1007/s00216-019-02184-4). The LLOQ in that article was 0.05 µgmL⁻¹. Therefore the paper cannot be accepted for publication in Royal Society Open Science.	Your effort and time in reviewing the manuscript are highly appreciated. While, regarding your valuable comment, The title of the article mentioned above looks close to the title of our work. However, the methodology, application, and data interpretation are different. The primary concern of our work, i.e. objective, is directed to design an analytical method capable of measuring the release of the new brand formulation. We added this article as a reference in the Introduction part. More details: [1] we developed a UPLC-MS analytical method. Parker et al, applied an HPLC-MS method using already marketed formulation as reference material. While we used reference standard materials throughout the work.

[2] The UPLC method, we used, showed better resolution within short run time, with improved throughput.

[3] Our method is applicable for quality control testing of drug release. The reported method could not be applied for testing the QC release because of the absence of reference standards.

[4] The reported procedure of sample extraction and analysis parameters are completely different from our procedure “different; HPLC column, elution mode, mobile composition, mass parameters (we used positive-isocratic elution, not alternative negative/positive switching MS)

[5] In the reported article, the calculated strengths were based on the instrumental response omitting the dilution factor. That is why the two values could not be matched.

[6] we have updated the literature review and added this article to our references, ref no. 14.